# Diversity and Spatial Distribution of Leaf Litter Curculionidae (Coleoptera: Curculionoidea) in Two Ecuadorian Tropical Forests

Oscar Maioglio [1], Cristiana Cerrato [2], Cesare Bellò [3] and Massimo Meregalli [4,*]

1    Associazione Naturalistica Piemontese, Museo Civico di Storia Naturale, Cascina Vigna, Via S. Francesco di Sales 188, 10022 Carmagnola, Italy
2    Biodiversity Monitoring Association—BioMA ETS, Via Pignari 20, 12037 Saluzzo, Italy
3    World Biodiversity Association Onlus c/o Museo Civico di Storia Naturale, Lungadige Porta Vittoria 9, 37129 Verona, Italy
4    Department Life Sciences and Systems Biology, Via Accademia Albertina 13, 10123 Torino, Italy
*    Correspondence: massimo.meregalli@unito.it

**Abstract:** Litter weevil communities were investigated in two different types of montane forest in Ecuador: foothill evergreen forest, present in the Otongachi Integral Reserve between 800 and 1000 m, and tropical montane cloud forest (TMCF), present in the Otonga Integral Reserve between 1600 and 2300 m. Sampling was conducted along elevation gradients, applying the entomological sifting method to 19 sampling transects (11 in Otonga and 8 in Otongachi). The taxa collected were identified as morphospecies, since the majority of them are still undescribed. A total of 510 specimens were sampled, belonging to 100 different morphospecies, 85 of which were found in Otonga and 15 in Otongachi. No species in common between the two areas were found, despite the fact that the distance is extremely small (approximately 12 km). The Otonga area, regardless of primary or secondary forest habitats, had higher mean species richness. In both areas, the majority of species were found in a small number of stations and were sampled in a small number of specimens. A dominant and ubiquitous species was present only in Otongachi. The analysis of the community composition was carried out according to different categories of environmental variables (Otonga vs. Otongachi, forest type, elevation, litter and canopy coverage characteristics). Two well-differentiated coenoses were found, influenced firstly by the differences between the two reserves, secondarily by the elevation gradient and, to a lesser extent, by the forest type and other environmental variables. Some morphospecies characteristic of a specific type of biocoenosis could be identified. The study highlighted how most litter weevil species are strictly associated with a peculiar microhabitat and have a very narrow elevation and spatial range; the weevil communities can be strongly affected by heavy human impact. The results confirm that the tropical forests, in particular the TMCFs, host great biodiversity and that the majority of species are strictly associated with a single site; therefore, particular protection for these habitats should be granted. However, the non-significant variation between primary and secondary TMCFs indicates that, when correct management is carried out, a limited disturbance can be tolerated with only a limited loss of biodiversity.

**Keywords:** tropical montane cloud forest; primary forest; secondary forest; biodiversity; forest conservation; forest management; weevils; litter arthropods

## 1. Introduction

Recent research on litter arthropods in some areas of Central and South America revealed the important role of weevils within the litter community of tropical forests, both in terms of their abundance, species richness and rate of endemism, and also as a possible indicator taxon of the biodiversity of tropical forests [1]. The weevils, along with the rove beetles, are the two most species-rich beetle taxa and are numerically dominant

within the litter community of tropical forests [1], thus constituting a key component in the biodiversity of these ecosystems and providing ecosystem services as decomposers of organic matter [2]. The communities of litter weevils are therefore useful to determine forest health and to evaluate the effect of forest management or damage to forest ecosystems [3] (and references therein).

Litter weevils can be considered good indicators of soil diversity in tropical montane cloud forests (TMCFs herein) as they reflect variations in species richness and community composition more accurately than other taxa that are not as well represented or are insufficiently diverse within these habitats [1]. They meet criteria that are used in the selection of litter bioindicator organisms [4], mainly that they are normally quite frequent in the targeted area, allowing collection of enough data for their analysis, and are usually characterised by low mobility and inability to fly, thus being closely tied to very circumscribed habitats. The litter weevils of TMCFs include numerous species that have a range of different ecological needs, making it possible to assess the causes of the presence or absence of species in the various survey areas and to recognise taxa indicators of specific coenoses. Importantly, the sampling techniques are simple, reproducible and can be easily standardised. Litter weevils are therefore suited for comparative studies between different sites.

TMCFs are one of the most threatened ecosystems in the world [5,6]. They globally occupy less than 380,000 square kilometres, which corresponds to 0.26% of the world's landmass. Of these cloud forests, 25.3% are found in tropical America, where, however, they account for less than 1.2% of the forested surface [7]. These forests are found in the tropical belt in mountainous areas, and are characterised by an almost continuous presence of clouds and fog, as moisture-laden air from the nearby oceans ascends towards the mountain ranges and, while cooling, it reaches the condensation point. TMCFs are extremely important sites of plant and animal diversity, with a high rate of endemism [8] (and references therein). In Ecuador, in the Cordillera Occidental (in which this research was conducted) the TMCFs are found in an elevation range between 1300 and 3000 m, most of them being located below 2600 m a.s.l. [9].

The purpose of our study was to gather semi-quantitative information on the litter weevil fauna of two Integral Reserves in Ecuador, Otonga and Otongachi. Particular effort was made to evaluate the composition of the litter weevil communities of the Otonga TMCF, selecting plots at various elevation levels and comparing patches in primary and secondary forest. We aimed at recognizing the degree of fidelity of each species to the type of forest and the elevation level, while also considering the response of this fauna to several other habitat variables. Samplings were also carried out in the degraded Otongachi secondary forest. In this case, other than comparing the weevil communities of the two reserves, we attempted to evaluate the utility of litter weevils as bioindicators of habitats with a high degree of naturalness as well as habitats severely impacted by anthropic activities. Changes in vegetation (determined by felling of forest areas, fires and other human activities) result in variation in the structure of the litter [10], which in turn influences, usually negatively, the composition of litter communities [11]. The different levels of impact given by disturbance on vegetation have been assessed through an approach that included soil arthropods as indicators of environmental quality [11,12].

*Current knowledge on the litter weevils of the Otonga region.*

Only few, scattered data are available, but no complete study of the Otonga weevil fauna has ever been attempted.

Voisin [13–18] described seven new species of Molytinae: Anchonini (*sensu* Voisin, 2013). Four of them belong to the genus *Baillytes* Voisin, 1992: *B. propinquus* Voisin, 1993, *B. loricatus* Voisin, 1996, *B. bartolozzii* Voisin, 1996 and *B. rotundicollis* Voisin, 1996. The other three species belong to three different genera, also new and simultaneously described by the author: *Boudinotes* Voisin, 2000, with *B. franciscanus* Voisin, 2000, *Capsonotus* Voisin, 2007, with *C. smilodon* Voisin, 2007 and *Mallanchonus* Voisin, 2009, with *M. lalisae* Voisin, 2009. All these species were recollected by one of the authors (C.B.) in more or less deep litter in TMCF.

Following the work of a few individual researchers from 1997 to 2002, but especially the research organised by the World Biodiversity Association (WBA) in 2004 and 2006, Bellò and Osella [19] described two new species of Cossoninae belonging to the genus *Howdeniola* Osella, 1980. One of these, *H. margheritae* Bellò & Osella, 2008 was found to be widespread in forest litter from 1500 to 2200 m in the Otonga TMCF.

Finally, Baviera, Bellò and Osella [20], from the material collected predominantly in the expedition organised by the WBA named "Ecuador 2008", reported for the first time the presence in Ecuador of the genus *Bordoniola* Osella, 1987, attributed to the subfamily Raymondionyminae, and described five new species, among which was *B. otongana* Baviera, Bellò & Osella, 2012, whose typical locality is the Otonga Integral Reserve, at 2000 m.

These data evidence that the knowledge of the Curculionoidea of Otonga is limited to only very few species, and the vast majority of the weevils native to the region are still undescribed.

## 2. Materials and Methods

### 2.1. Study Area

2.1.1. Otonga Integral Nature Reserve

The Otonga Integral Nature Reserve (Figure 1) is located at an elevation between 1300 m and 2300 m a.s.l. It lies on the border between the provinces of Pichincha and Cotopaxi, about 4.5 km west of San Francisco de las Pampas (Canton of Sigchos, province of Cotopaxi) with coordinates 00°25′ S, 79°00′ W. It extends for about 1500 hectares and includes approximately 1000 hectares of primary forest, while the remainder consists of naturally regenerating pastures and areas of secondary forest planted since the 1990s by nurseries in the same reserve; the tree species that were planted are all native. The forests belong to two different coenoses, both attributed to the Chocò-Darien Moist Forest ecoregion [21,22]: the low-elevation evergreen mountain forest (EMF herein), between 1300 and 1700 m, and the TMCF in the higher part [12]. The former is found in the lowest part of the reserve, and is developed especially on the slopes of the Rio Las Damas. It is a forest with trees between 25 m and 30 m tall and hosts most of the species characteristic of low-elevation forest areas that, in some cases, reach the upper limit of their range (Myristicaceae). TMCF is a forest with trees between 20 m and 25 m tall, rich in mosses and epiphytes such as Bromeliaceae and Orchidaceae, with many species. Bamboos have a very high diversity in this habitat, with 45 out of the 54 species known for Ecuador [23]. The climate is humid and relatively warm, characterised by the presence of fog and persistent clouds. A weather station was placed in Otonga from May 1996 to April 1997, at an elevation of 2000 m, which made it possible to collect climatic data: the mean annual humidity is 90%, the mean annual temperature is 16 °C and the mean annual rainfall is over 2500 mm [24]. Although the temperature is always between 15 °C and 27 °C, and rainfall is abundant throughout the year, there are slight variations in these two values, resulting in two different seasons: the rainy season from December to June, and the dry season from July to November [25].

2.1.2. Otongachi Reserve and Environmental Education Center

The Otongachi Reserve and Environmental Education Center are located less than 1 km from La Union Del Toachi, Pichincha Province, in the foothills of the Cordillera Occidental, at an elevation of 800 to 1000 m a.s.l. Otongachi officially belongs to the "Reserva de Bosque Integral Otonga" and is about 12 km north of Otonga (00°23′ S, 78°58′ W). It covers a total of about 150 hectares. Within this area is a secondary foothill evergreen forest (FEF herein) [12], a botanical garden that houses most of the plant species typical of this elevation in Ecuador (many of which are useful to humans as edibles, officinal or used in local industry and handicrafts) and a centre for environmental education. The acquisition of this reserve by Fundación Otonga follows that of Otonga. It has primary importance both for the habitat it protects and for its positioning along an important communication road that connects villages in the vicinity. The role this reserve plays is as a reference point for the local community and as a support point for scholars and researchers carrying out

expeditions to areas of Ecuador that are more isolated and challenging to reach. Otongachi lies along a slope that does not exceed 1000 m a.s.l., near the confluence of the Rio Toachi and Rio Pilaton, which flow respectively to the east and north of the Reserve. The part of the Otongachi Reserve lying close to the Toachi River has quite steep slopes. The climate is hot and humid and is characterised by a mean annual temperature between 18 °C and 24 °C, with the lowest temperatures higher than those recorded in Otonga, and a mean annual rainfall between 1000 and 2000 mm [26]. As in Otonga, in Otongachi there is also a dry and a rainy season, in the same periods of the year. The proximity to the large forest corridor formed by the Ecological Reserve "Los Illizinas", the Otonga Integral Reserve and the Rio Lelia Forest Reserve allows Otongachi to maintain a high diversity, constituting an important refuge for the flora and fauna throughout the region [27–29]. However, studies on the more highly specialised taxa, such as scarcely mobile litter arthropods, are as yet very incomplete and do not offer detailed information.

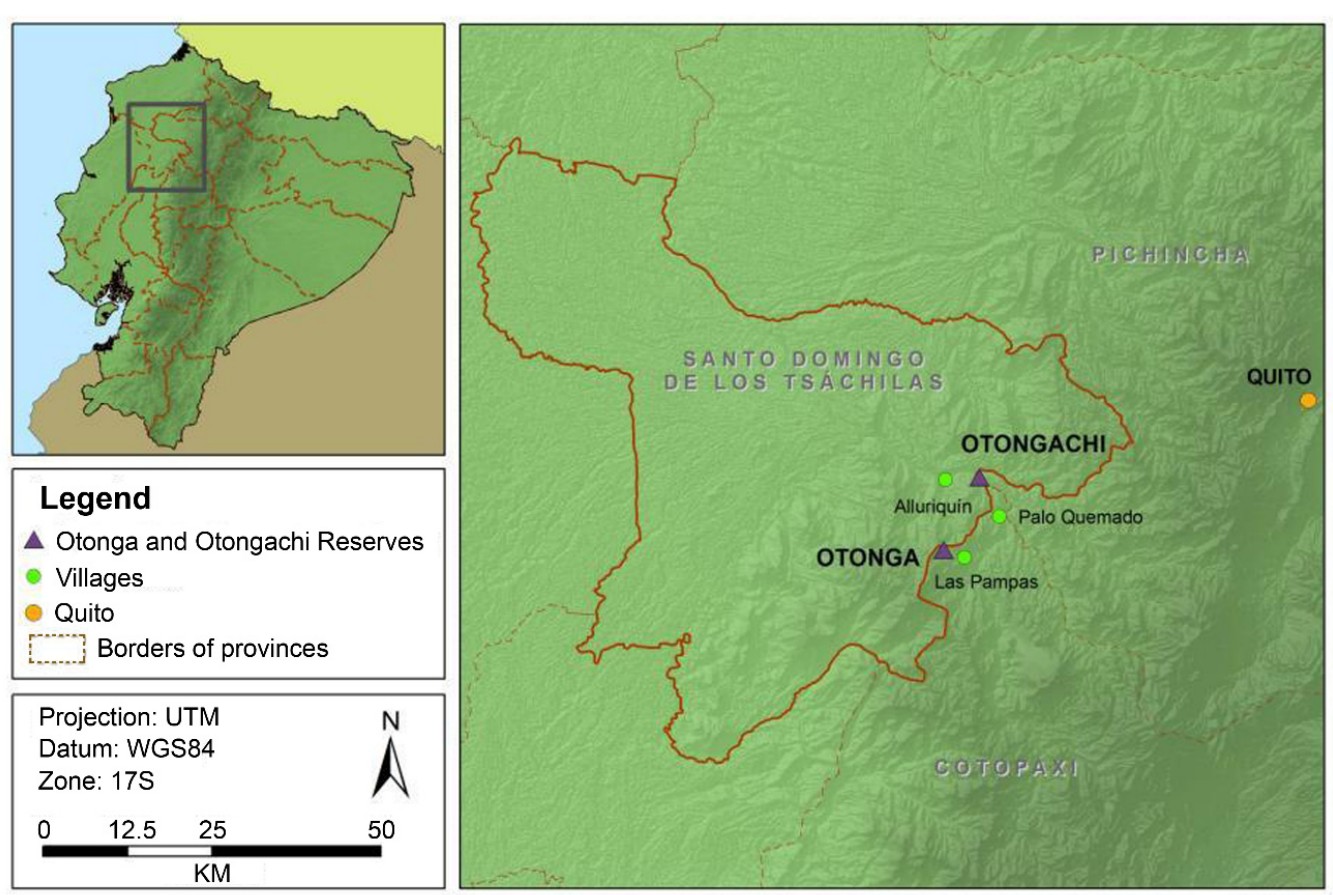

**Figure 1.** Localisation of the study area.

### 2.2. Data Collection

A field survey was conducted between April and May 2014. The areas suitable for the sampling were selected based on the following criteria: 1. Accessibility—sample points located no more than 20 m from the main forest trail; 2. Presence of soil litter and of a rich organic substrate; 3. Absence of anthropogenic disturbance—e.g., no roads, buildings or grazing animals; 4. Absence of natural disturbance—e.g., soil erosion and landslides, areas subject to flooding or too close to water basins. Both primary and secondary forested areas were investigated in Otonga while only secondary forest, the only type existing, was

sampled in Otongachi. EMF in Otonga, differently from what was originally planned, was impossible to access.

Each sample followed a protocol of linear transects distributed over an elevation gradient. In order to differentiate the samplings according to the elevation, levels with a range of 100 m were chosen. In Otongachi, two elevation levels were sampled (elevation ranges of 800/900 m and 900/1000 m), located on the orographic right of the Rio Toachi. In Otonga, six elevation levels were selected, from 1700 to 2300 m, located on the orographic left of the Rio Esmeraldas. For each elevation level, two sampling plots were chosen: one in an area of primary forest and one in an area of altered secondary forest. In a few cases, it was necessary to vary the number of sampling plots. In Otonga, the highest elevation level (2200–2300 m) is located on top of the highest mountain and is characterised by a very narrow area completely covered with well-preserved primary forest: in this case, only one sampling plot (primary forest) was selected. In Otongachi, four sampling plots were selected per elevation level. Since there is no primary forest and the elevation range of the area investigated is only 200 m, we decided to duplicate the samplings at each elevation level, so as to have a number of transects comparable to those of Otonga. A total of 19 transects, 8 in Otongachi and 11 in Otonga, were sampled by one of the authors (O.M.) (Figure 2).

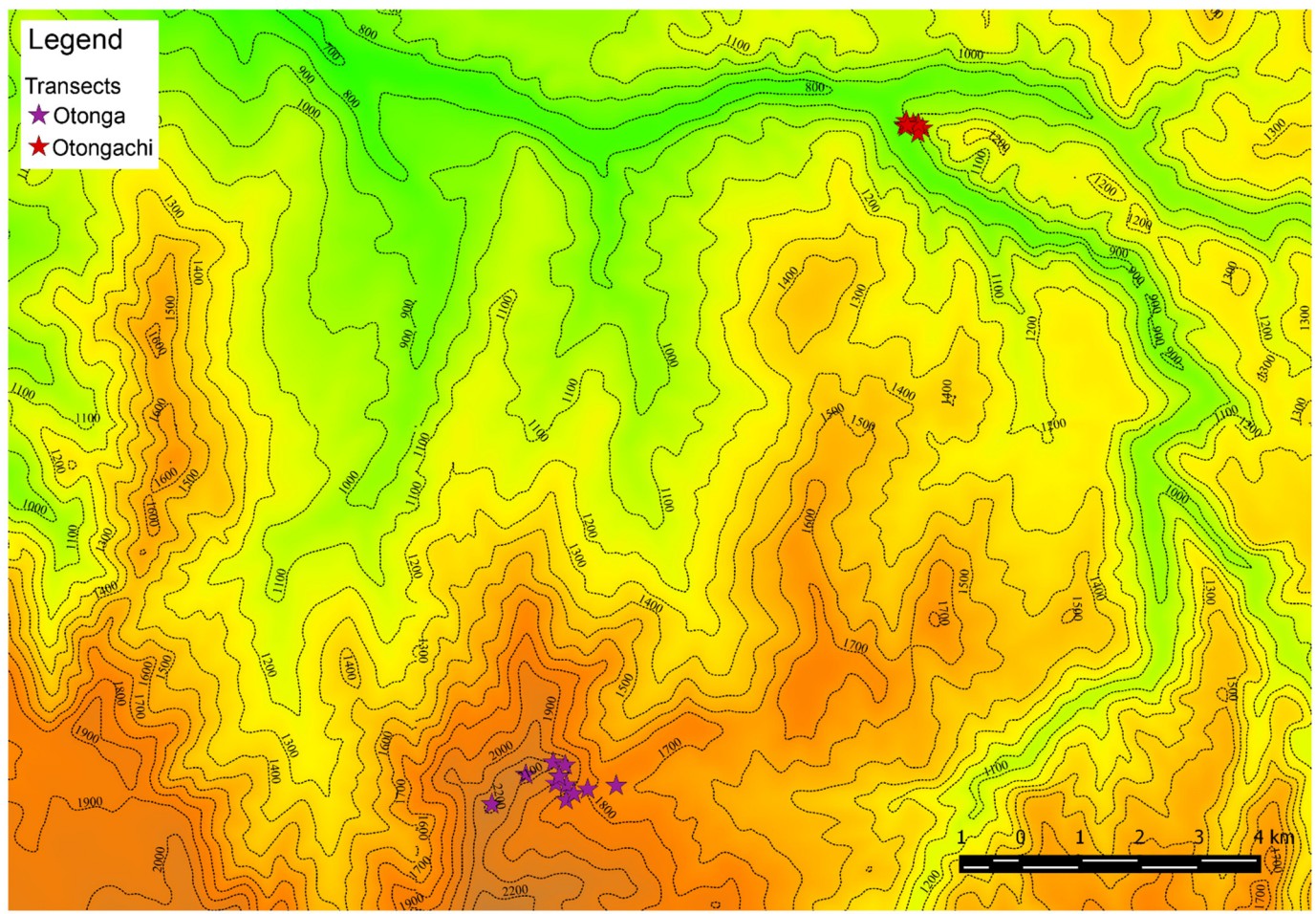

**Figure 2.** Localisation of the transects.

Each transect is the result of five litter samplings, collected in areas of one square metre (squares of 1 m × 1 m), 50 metres apart from each other. The site on which a transect was started was identified by checking the elevation with GPS and the type of habitat as indicated by a naturalist guide who assisted during the samplings; whenever possible, the

one-square-metre areas were chosen to include parts of decaying wood in areas adjacent to large trees. Each sample consisted of a bag of screened litter obtained from the five one-square-metre areas present in a transect. The first step was to grind with a machete the litter present on the ground and to collect the organic fraction of the litter by placing it in an entomological sieve with a 1 cm mesh size. The operation of collecting, sorting and screening the litter on each one-square-metre area was repeated until reaching a bucket capacity of 3 L of refined litter. In cases where, within the one-square-metre area, the litter was particularly abundant and not completely homogeneous (due to a considerable difference between the surface fraction and the deeper fraction, or in the presence of decaying wood only in one part), the various components were subsampled equally, and placed in the sieve. After the sampling of the 5 one-square-metre areas, each transect consisted of 15 L of sifted soil.

Previous studies on litter weevils conducted in TMCF [1,8,20] have shown that good results can be obtained with dynamic extraction methods, involving the use of Winkler and Berlese sorters. In our study, twenty Winkler sorters and eight Berlese sorters were used; for each 15-L sample of litter, 12 L were extracted in a Winkler sorter and 3 L in a Berlese sorter. For the Winkler sorter, after 5 days since the first day of extraction, and again every 5 days thereafter, the mesh bags were extracted and the litter inside them was remixed by hand. Each sample placed in the Winkler sorters underwent an extraction process lasting no less than 15 days. The Berlese sorters were placed under a 40 W lamp. After 24 h since the start of the extraction, and again every 24 h thereafter, the litter in the Berlese sorters was extracted and remixed by hand. Each sample placed in the Berlese sorters underwent an extraction process lasting no less than 4 days. Specimens were conserved in a mixture of 95% concentrated ethyl alcohol and white vinegar, in a 3:2 ratio, until preparation on entomological cards.

In areas where biodiversity is particularly high but poorly studied, the analysis and identification of specimens of various groups of arthropods can present difficulties even when performed by specialists, who often encounter what has been termed "taxonomic impediment" [30], a problem determined by the majority of taxa being undescribed [31]. As previously mentioned, Otonga's weevil fauna is scarcely known, and a large proportion of the specimens found in the course of the present study belong to new species. A traditional classification into genera and species would therefore require a lengthy work with preliminary descriptions. To obviate this problem, the study of the specimens collected was carried out by grouping them into morphologically uniform taxa, that is, morphospecies, each of which was identified by an alpha-numerical code, which was assigned according to the following scheme: at first, each taxon was roughly assigned to a morphogenus, labelled with a letter; subsequently, within each morphogenus, each different taxon was labelled with a progressive number. Practically, each morphospecies corresponds to a binomial species in the sense of traditional taxonomy. This method, with some differences according to the knowledge of the taxa in the area investigated, termed "parataxonomy" [32], was used by various authors i.e., [1,8,33] in studies on weevils, rove beetles and carabids. Identification of morphospecies was based on external characters, and genitalia were dissected and examined only in cases of uncertain interpretation of very similar discriminating characters. In this paper, to simplify, the terms genera and species will be used for morphogenera and morphospecies, respectively.

### 2.3. Data Analysis

The analyses were performed with R 4.2.1 software [34]. Values, where not otherwise specified, are given as mean ± standard error.

### 2.3.1. Analysis of Topographic and Environmental Variables

The following topographic variables were calculated for each plot with Excel data tables and GIS software [35]: mean, lowest, highest elevation and coordinates of the midpoint. For the characterisation of the data, we started from an elevation digital model

(DEM 50, source: [36]) with a 50 m × 50 m resolution. Each plot was assigned to a forest type (primary or secondary) and to one of the two reserves (Otonga or Otongachi). The habitat type attributed to the plots was TMCF for those located in Otonga and FEF for those located in Otongachi. Each plot was also assigned to one of the 100 m elevation levels identified along the elevation gradient considered (from 1700 to 2300 m in Otonga, and from 800 to 1000 m in Otongachi).

The data collected at each single sampling point of a plot were analysed to determine the habitat variables for rotting wood, moss, litter moisture and canopy coverage. Data for these variables were gathered from field observations and photographs of the sampling location and the canopy, and a portable soil moisture detector (Fosmon Soil Tester Meter).

The presence of decaying wood (indexed as rotten_log), moss (moss_index) and moisture in the litter (dryness_index) at each of the 5 sampling points of a plot was categorised with a value varying from 0 to 1; the value 1 was assigned to a high presence of decaying wood, moss and dry litter, while the value 0 was assigned to the opposite cases. The values assigned to these indices were obtained from a reclassification of the categories created for each plot by summing the values of the 5 points of each plot:

- rotten_log and moss_index. An index with a value of 0 represented a plot with the sum of the values of the 5 sampling points between 0 and 2 (indicating the scarcity of the variable considered within the plot), while an index with a value of 1 represented the plot with a sum of the values of the 5 sampling points between 3 and 5 (indicating the abundance of the variable considered within the plot).

- dryness_index. An index with a value of 1 represented a plot with the sum of the values of the 5 sampling points between 0 and 1; an index with a value of 2 represented a plot with the sum of the values of the points between 2 and 3; an index with a value of 3 represented a plot with the sum of the values of the points between 4 and 5.

The canopy coverage was indexed according to the mean percentage of the coverage detected at each of the 5 sampling points of the plot considered.

The rotten_log, moss_index, dryness_index and canopy coverage were then used to compare the plots.

### 2.3.2. Estimators and Rarefaction Curves

As this type of sampling cannot be exhaustive, a number of non-parametric estimators of species richness were used in addition to the number of species observed (S.Obs), to obtain an overall estimate of the number of taxa expected in each of the plots located in Otonga and Otongachi. The abundance-based coverage estimator (ACE), Chao1, Jacknife1 and Jacknife2 estimators were used. ACE is a simple estimator of the number of species in a sample (abundance) that considers both rare and common species, i.e., those present in at least ten samples. The other estimators use the distribution of the rare species to estimate the community species richness. These models are based on the assumption that the probability of capture of rare species increases as the number of samples increases. A variance calculation was performed to assess the reliability of the Chao1 estimate. Chao estimates are suitable in cases where there is a dominance of relatively rare species, such as in inventories of groups of hyperdiverse arthropods [37], so they are appropriate for litter weevil communities [1]. To have a comparison, the incidence estimators were also used. The accumulation curves of each plot in the two study areas and their respective rarefaction curves were calculated, using data from the Chao1, Jacknife1, Jacknife2 and ACE samplers and estimators. The rarefaction curves were then represented on a graph correlating sample size and species richness per sampling station (sample size).

### 2.3.3. Analysis of Species Richness

The species richness of the communities was analysed in Otonga and Otongachi, in the different forest types (primary and secondary) and in the different elevation ranges. A count of the specimens and species collected was carried out, both as the total number of taxa sampled in the two reserves and the taxa sampled in each of the two reserves, considered

individually. The mean numbers of species and specimens per plot were calculated and a comparison among the various plots was made. The quantity and frequency of "singletons" (only a single specimen of a species found in one plot) and "doubletons" (two specimens) sampled in each of the two reserves and in both forest types were pinpointed, and the species shared between the primary and the secondary forest, or present in only one of them, were quantified. The number of plots in which each species was found was counted, highlighting those present in a high number of plots and noting their distribution within different forest types or different elevation belts.

In order to highlight significant variations in the species richness, the following statistical tests were implemented, applied on data of abundance and presence–absence of species in the sampling plots.

- Kruskal–Wallis test (KW herein): the KW test revealed differences in species richness between plots by assessing the influence of variables with more than two categories, such as dryness_index and canopy coverage.
- Mann–Whitney test (MW herein): the MW test showed differences in species richness between plots by evaluating variables that can be divided into two categories, such as moss_index, rotten_log, forest type (primary and secondary) and reserve type (Otonga and Otongachi).

In order to identify potential variations in the relative frequencies of the species along the elevation gradient, the MW test was again used to test for differences in community composition between elevation belts. The same tests were also conducted on the rare species, first starting with the singletons and doubletons in each individual plot (considering one plot at a time), and secondly by taking into account the rare or infrequent species (rare_num, found in only 1 or 2 specimens; rare_freq, found only in 1 or 2 plots). The correlation between the species richness and the number of singletons, doubletons and rare species for each plot was tested using Spearman's rank correlation index and Pearson's correlation index. The same indices were used to assess possible correlations between the species richness and the percentage values of the different rarity categories considered, such as: the percentage of species that are uncommon in the plots (rare_freq), the percentage of rare species in each individual plot and the percentage of rare species collected over all the samplings (rare_num).

### 2.3.4. IndVal Analysis

The IndVal method [38] was used to recognise species indicative of a particular plot and/or forest type within the Otonga and Otongachi Reserves. This method involves the use of a quantitative index capable of measuring the degree of association of a species with any group, defined as a grouping of a series of sampling stations, or sites. The index is maximum when the species is found only within the group considered and every site in the group contains the species. We tested for the association of each species of weevils according to the forest types (primary and secondary) and the two reserves. It was thus possible to identify species indicative of a forest type (primary or secondary) and species indicative of one of the two areas. This approach allowed the identification of species potentially useful as indicative of coenoses related to the categories considered. The IndVal method was applied using matrices of presence–absence, through the "indicspecies" package [39] using 999 permutations. Only the combinations of forest type–area that had an ecological significance were considered in the permutations. IndVal values greater than 0.7 and a *p*-value of less than 0.05 were considered significant.

### 2.3.5. PCoA

Principal coordinate analysis (PCoA) was used for a qualitative analysis of the community composition. The use of PCoA allowed the sorting of the starting dataset in such a way as to visualise the differences between the communities present in the sampling sites (plots) between Otongachi and Otonga, based on abundance and presence/absence data.

2.3.6. Community Composition—PERMANOVA Analysis

To highlight the significance of any differences in the communities in the plots analysed, the non-parametric multivariate analysis of variance (PERMANOVA) was used, based on distance matrices calculated according to the Jaccard index. The analysis was conducted using the "adonis" function of the "vegan" package [40]. PERMANOVA uses a statistic referred to as the pseudo F-ratio in the analysis of sample variance. A high value indicates that sites in a particular grouping (i.e., Otongachi) are closer to each other within the multivariate space than they are to another grouping (i.e., Otonga).

The significance in the *p*-values obtained through PERMANOVA in the composition of communities may be due to a real difference in the groupings within the multivariate space, or to a difference in the dispersion of communities in the multivariate space within the grouping itself [41]. In order to highlight which of the two factors is responsible for the level of significance obtained, multivariate dispersion analysis using permutations was employed (PERMDISP). In this case, the analysis was conducted using the "betadisper" function of the "vegan" package [40]. The analysis involved calculating the centroid of each grouping and thus the distance of each plot from the grouping centroid. An analysis of variance or permutational ANOVA was then performed to calculate the mean values of the dispersion between the groupings. A pseudo F-statistic and a *p*-value were then calculated in the same way as described for the PERMANOVA. The difference in the composition of the weevil communities was tested against the Otonga and Otongachi Reserves.

2.3.7. Spatial Analysis

Data on community composition were examined in a variation partitioning context, as proposed in [42,43]. This methodology allows the analysis of groups of non-independent variables and the identification of the portion of variability in the data that is uniquely explained by the different groups of variables. The explanatory variables, measured at plot level, were assembled into 4 groups: elevation (expressed for each plot by considering the value of the upper limit of the 100 m elevation interval in which it was collected); area (Otonga or Otongachi, as a categorical variable); forest (primary or secondary, as a categorical variable); habitat (expressed for each plot by the sum of the values of dryness_index, rotten_log, moss_index and canopy).

The variables of the plots considered in the analyses can be found in the Supplementary Material Table S1.

Variation partitioning analysis was carried out following 4 basic steps.

(i) Selection and identification of the explanatory variables within each of the 4 groups.

(ii) Analysis of each group of variables through the application of canonical sorting techniques. Through the application of db-RDA to the weevil matrices, the variability in the data explained by each group of predictor variables was calculated and expressed as a percentage of adjusted $R^2$adj [42,44].

(iii) Calculation of fractions of interest, so as to identify the full amount of variability explained by each group of variables (total effect) and the variation explained independently by each group, conditional on the other variables (pure effect).

(iv) Calculation of the significance of the individual fractions by randomisation. In each iteration, the response variable is subjected to a randomisation procedure, whereby the order of the samples is changed. In each randomisation, the variation is partitioned into its components. The operations are performed 999 times, such that the frequency distribution of the corrected $R^2$adj value is obtained, under the null hypothesis of a random association between the response and the explanatory variables. The *p*-value is calculated as the probability of obtaining a value of $R^2$adj greater than or equal to the observed value.

**3. Results**

*3.1. Habitat Variables*

The habitat variables collected at each of the five points along each transect were merged and led to the following results (Supplementary Material Table S1).

The rotten_log index shows more transects with index = 1 in Otonga (8 plots out of 11) than in Otongachi (2 plots out of 8). There is no substantial difference between the two types of forest in Otonga: in the primary forest the plots with rotten_log index = 1 are 4 out of 6, while in the secondary forest they are 4 out of 5.

The moss_index shows a slightly higher number of plots with index = 1 in Otonga (5 plots out of 11, all in primary forest) compared to Otongachi (3 plots out of 8). There are no plots with moss_index = 1 in the secondary forest of Otonga.

Litter is generally less humid in Otonga (dryness_index = 3 for 8 plots out of 11) than in Otongachi (dryness_index = 3 for 1 plot out of 8), where more plots with a low dryness_index were found (dryness_index = 1 for 5 plots out of 8). No substantial differences were seen between primary and secondary forests in Otonga.

The comparison of canopy coverage from the plots in Otonga and Otongachi primary and secondary forests shows no particular differences, with mean coverage ranging from 56.3% in Otonga primary forest (56.6% in secondary forest) to 59% in Otongachi.

*3.2. Species Richness*

A total of 510 specimens of Curculionoidea, identified as belonging to 100 different morphospecies, included in 24 different morphogenera, were collected (Supplementary Material Table S2). The species considered as typical of the soil litter share several functional traits. All of them are apterous, usually small, microphthalmic or even anophthalmic, and most of them have integument with broad setae, tubercles and granules. Even though functional traits of weevils associated with forest litter have never been evaluated, this combination of characters is often found in these taxa, in tropical as well as temperate forests (M.M., unpublished data). Very few winged, macrophthalmic species were sampled, and these were excluded from the analyses, since they were considered to be "tourists", casual findings of species not normally associated with the habitat investigated. If added to the analyses, they could interfere with the results. In the 11 plots of Otonga, 393 specimens belonging to 85 morphospecies were collected, while 117 specimens belonging to 15 morphospecies were collected in the eight plots of Otongachi. The majority of species were Molytinae: Anchonini; other taxa belonged to Raymondionyminae, and in particular to the genus *Bordoniola* Osella, 1987. However, we did not attempt at any formal identification, which may be carried out later.

Rarefaction curves were calculated for all transects monitored and then represented on a graph correlating sample size and morphospecies richness per sampling station. Looking at the graph, it can be seen that the entire estimated community was never sampled, as no curve reaches the asymptote (growth close to zero as the number of specimens sampled increases). A markedly different pattern in the development of the rarefaction curve can be recognised for the plots in Otonga. These have faster growing curves and show a higher mean richness of species per plot, even in the case of relatively small sample sizes, than the plots in Otongachi. In particular, the Otonga plot "e270" has the fastest growing rarefaction curve. The only exception for Otonga is plot "e260", which shows a rarefaction curve that is intermediate between those of Otonga and those of Otongachi (Figure 3).

The estimated species richness was calculated for each plot as the mean of the corresponding values obtained from the Chao1, ACE, Jacknife 1 and Jacknife 2 estimators (Table 1).

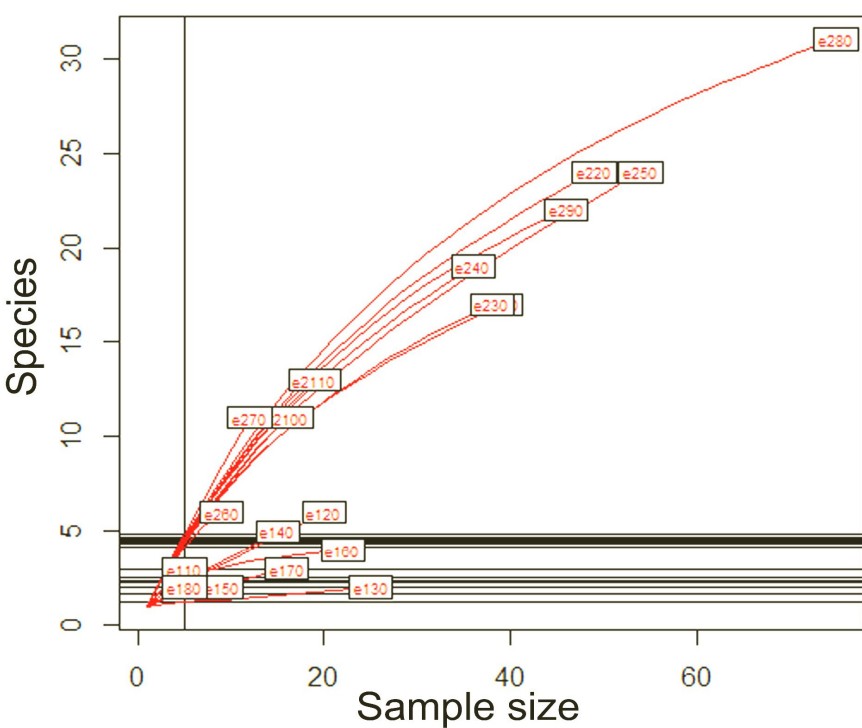

**Figure 3.** Rarefaction curves. Sample size corresponds to the number of individuals sampled; codes in the boxes are the codes of the sampling plots.

**Table 1.** Observed species richness (S.Obs) and estimators Chao1, ACE, Jacknife1 (Jack_1), Jacknife2 (Jack_2), and the mean value among all estimators. NA = value of the estimator not obtained.

| Estimators / Plots | S.Obs | Chao1 | ACE | Jack_1 | Jack_2 | Mean |
|---|---|---|---|---|---|---|
| e110 | 3.00 | 4.00 | 6.67 | 4.60 | 4.60 | 4.97 |
| e120 | 6.00 | 16.00 | NA | 10.75 | 14.50 | 13.75 |
| e130 | 2.00 | 2.00 | NA | 2.96 | 3.76 | 2.91 |
| e140 | 5.00 | 11.00 | NA | 8.73 | 11.40 | 10.38 |
| e150 | 2.00 | 2.00 | 2.00 | 2.00 | 2.00 | 2.00 |
| e160 | 4.00 | 4.00 | 5.11 | 4.95 | 4.86 | 4.73 |
| e170 | 3.00 | 4.00 | NA | 4.88 | 6.25 | 5.04 |
| e180 | 2.00 | 2.00 | 3.13 | 2.80 | 2.80 | 2.68 |
| e210 | 17.00 | 29.00 | 27.50 | 25.77 | 31.77 | 28.51 |
| e2100 | 11.00 | 20.33 | 29.47 | 18.50 | 22.37 | 22.67 |
| e2110 | 13.00 | 25.00 | 27.36 | 21.53 | 26.47 | 25.09 |
| e220 | 24.00 | 43.50 | 40.04 | 36.73 | 45.59 | 41.47 |
| e230 | 17.00 | 29.00 | 29.91 | 25.76 | 31.74 | 29.10 |
| e240 | 19.00 | 30.00 | 33.53 | 29.69 | 35.50 | 32.18 |
| e250 | 24.00 | 50.25 | 54.90 | 38.72 | 49.50 | 48.34 |
| e260 | 6.00 | 6.75 | 9.00 | 8.67 | 7.96 | 8.09 |
| e270 | 11.00 | 33.50 | 66.00 | 20.17 | 25.24 | 36.23 |
| e280 | 31.00 | 40.75 | 43.45 | 43.83 | 49.24 | 44.32 |
| e290 | 22.00 | 28.43 | 31.99 | 31.78 | 35.08 | 31.82 |

The graph shows that the observed species richness is the result of an underestimation in all areas sampled in both reserves, and the gap with the estimated species richness value is most pronounced (it reaches a value greater than or equal to twice the observed value) in two plots in Otongachi ("e140", "e120") and in three plots in Otonga ("e250", "e270", "e2100").

The scatter plot and Spearman's correlation coefficient (0.953, $p < 0.001$) show, however, that there is a positive and significant correlation between the observed and estimated values of species richness (Figure 4).

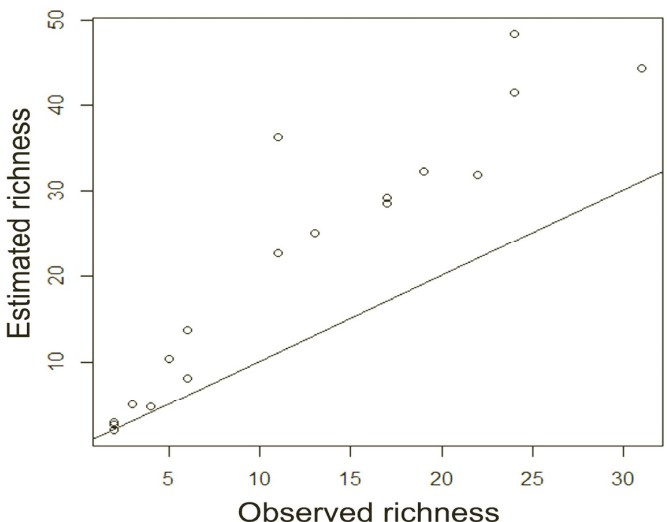

**Figure 4.** Observed and estimated species richness. The empty circles correspond to the sampling plots.

The abundance and diversity indices (Shannon and Simpson) were found to be positively and significantly correlated with species richness (Spearman, S-N, $\rho = 0.798$, $p < 0.001$; S-Shannon, $\rho = 0.974$, $p < 0.001$, S-Simpson, $\rho = 0.931$, $p < 0.001$), so that only species richness was considered in the subsequent analyses. Significant differences in mean species richness per plot were observed between the Otonga and Otongachi Reserves. The mean abundance of species and the mean number of specimens per plot are higher in Otonga ($17.7 \pm 2.2$ species per plot, with $35.7 \pm 6.1$ specimens) than in Otongachi ($3.4 \pm 0.5$ species per plot and $14.6 \pm 2.7$ specimens) (MW test: V = 87.5, $p$-value < 0.01) (Figure 5).

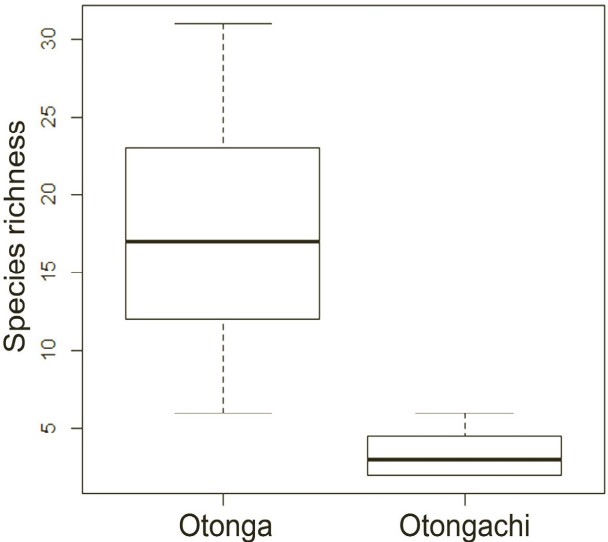

**Figure 5.** Differences in species richness per plot between Otonga and Otongachi.

The difference between the species richness per transect between the primary and secondary forests, considering both reserves, was also significant (MW, V = 63, *p* = 0.04), with a *p*-value higher than that obtained from the reserve comparison. However, these results were essentially determined by the differences in species richness between Otonga and Otongachi. No significant difference emerged, in fact, in species richness between primary and secondary forest in Otonga (mean number of species per plot 18.0 ± 3.7, with 36.5 ± 10.0 specimens in the primary forest, 17.4 ± 7.5 species with 34.8 ± 7.5 specimens per plot in the secondary forest, MW: V = 15.5, *p* = 1.0). There was no significant difference in the mean species richness per plot observed in any of the following cases: between plots with high or low moss_index (MW: V = 31, *p*-value = 0.3); between plots with high or low rotten_log index (MW: V = 26, *p*-value = 0.13); between plots with different litter moisture (dryness_index, KW: $\chi^2$ = 5.34, df = 2, *p*-value = 0.07); between plots with different canopy coverage (KW: $\chi^2$ = 3.15, df = 3, *p*-value = 0.37). A significant correlation was found between elevation and species richness (Spearman's correlation, ρ = 0.895, *p* < 0.001). Species richness increased as a function of elevation, apparently more markedly above 1700 m, although information on species richness at intermediate elevations between Otongachi and Otonga would be needed to confirm this finding (Figure 6).

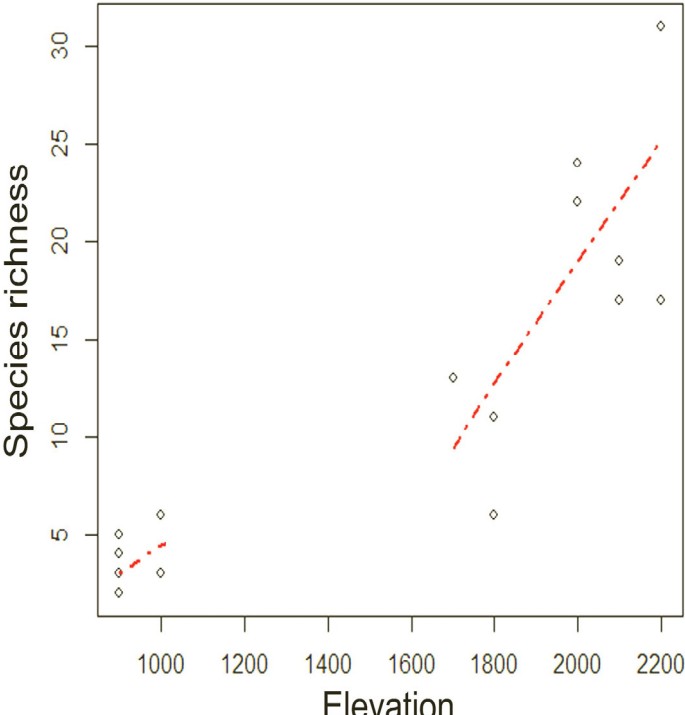

**Figure 6.** Transect species richness as a function of altitude. In red, the LOWESS regression line, which describes in a non-parametric manner the trend in species richness.

Within each plot, the percentage trend of singletons and doubletons was analysed as a function of environmental variables. The number of rare species was related to the total species richness (ρ = 0.990, *p* < 0.001), but not their percentage trend (ρ = 0.300, *p* = 0.212). The relationship between the proportion of rare species and the variables characterising the sampling stations yielded the following results: differences at the limits of significance between the Otonga and Otongachi Reserves (MW, V = 67, *p*-value = 0.062), with mean values higher in Otonga; no difference between primary and secondary forests, considering both Otonga and Otongachi (MW, V = 49.5, *p*-value = 0.378); no difference between primary and secondary forests, considering only Otonga (MW, V = 14.5, *p*-value = 1); no difference between plots with high or low moss_index (MW, V = 40, *p*-value = 0.77); no difference between plots with high or low rotten_log index (MW, V = 36.5, *p*-value = 0.51);

no difference between plots with different litter moisture index (KW, $\chi^2$ = 0.12, df = 2, $p$-value = 0.94); no difference between plots with different canopy coverage (KW, $\chi^2$ = 2.94, df = 3, $p$-value = 0.40). There is no clear linear relationship in the proportion of singletons and doubletons per transect as a function of elevation (Spearman's correlation, $\rho$ = 0.209, $p$ = 0.391). There is a growth in the percentage of rare species per plot directly proportional to elevation in Otongachi, and a growth inversely proportional to elevation in Otonga, but since Otongachi extends only for two elevation levels, the difference in percentage may also be determined by random factors due to non-exhaustive sampling.

Rare species over the totality of the samplings were classified according to two rarity criteria: low abundance (rare_num) and low frequency (rare_freq). These values, expressed as a percentage of plot species richness, were found to be positively correlated with each other (Spearman correlation, $\rho$ = 0.981, $p$ < 0. 001), but neither with plot species richness (rare_num, $\rho$ = 0.105, $p$ = 0.668; rare_freq, $\rho$ = 0.187, $p$ = 0.444), nor with the percentage of singletons and doubletons (rare_num, $\rho$ = 0.358, $p$ = 0.132; rare_freq, $\rho$ = 0.278, $p$ = 0.250). No significant difference for these two categories of rare species was observed either in the comparison between study areas, or between the two forest types, or as a function of environmental variables. The only case in which significant differences were observed is as a function of the dryness_index (KW: rare_num, $\chi^2$ = 7.576, df = 2, $p$ = 0.023; rare_freq, $\chi^2$ = 8.984, df = 2, $p$ = 0.011), indicating that intermediate values of litter moisture have a lower percentage of rare species.

In none of the following cases was a significant influence of environmental variables on the percentage of rare morphospecies observed: between the Otonga and Otongachi Reserves (MW, rare_num, V = 39.5, $p$-value = 0.740; rare_freq, V = 40.5, $p$-value = 0.804); between primary and secondary forests (MW, rare_num, V = 41, $p$-value = 0.895; rare_freq, V = 45.5, $p$-value = 0.598); between these two forest types in Otonga (MW, rare_num, V = 18.5, $p$-value = 0.582; rare_freq, V = 22, $p$-value = 0.247); between plots with high or low moss_index (MW, rare_num, V = 39, $p$-value = 0.709; rare_freq, V = 33.5, $p$-value = 0.408); between plots with high or low rotten_log index (MW, rare_num, V = 51, $p$-value = 0.652; rare_freq, V = 51, $p$-value = 0.653); between plots with different canopy coverage (canopy, KW, rare_num, $\chi^2$ = 4.8, df = 3, $p$-value = 0.187; rare_freq, $\chi^2$ = 5.112, df = 3, $p$-value = 0.164).

A variation partitioning analysis was applied to assess the relative roles of the different groups of variables (elevation, study area, forest type and microhabitat characteristics) in determining the observed patterns of species richness. The following variables were analysed: the partitioning of the variability of species richness by transect; the partition of the variability of the number of rare species per transect; the partition of the variability of the percentage of rare species per transect. The percentage of total variability explained by the model consisting of the variables considered was: 75% on the data of species richness per transect; 79% on the data of number of rare species per transect; 36% on the data of percentage of rare species per transect. In two out of three models (species richness and number of rare species), most of the variability was determined by the elevation factor. In the third model (percentage of rare species), the variability was determined by the area factor, when considered in relation to the other variables. The second element in terms of explanatory importance is the area factor in two out of three models (species richness and number of rare species) and the elevation factor in the third model, when considered as a function of the other variables. The categories "forest" and "habitat" almost entirely lose their explanatory power when conditioned to the other variables. All these components present overlaps, as can be seen in Figure 7. In most cases, the elevation component is explanatory of a greater percentage of variability than the area component.

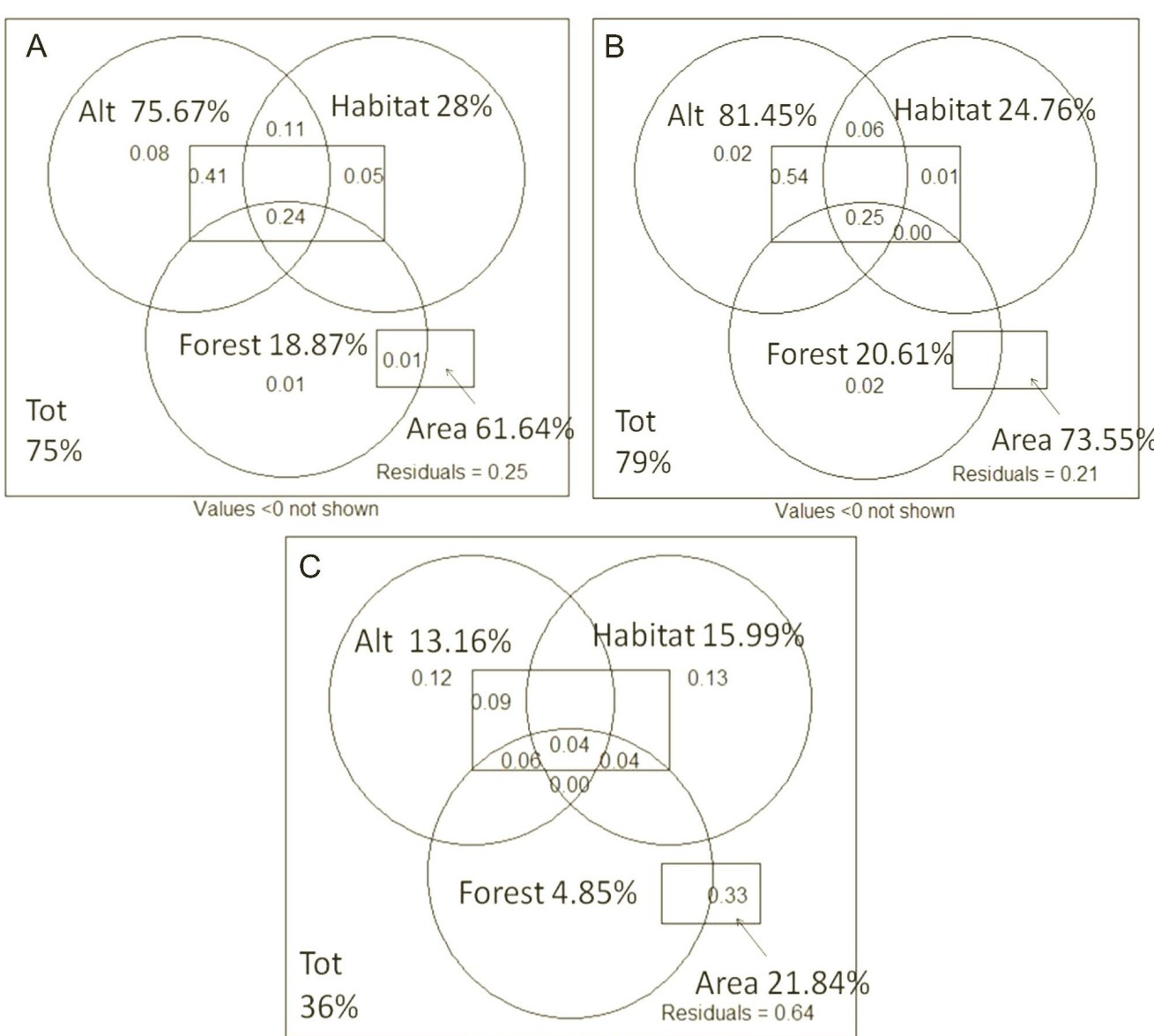

**Figure 7.** Percentage of variability explained by the four indicated components. Forest = primary or secondary forest, Alt = elevation levels, Area = Otonga or Otongachi, Habitat = other environmental variables. Models built on data of (**A**) species richness per plot; (**B**) number of rare morphospecies of each individual plot; (**C**) percentage of rare morphospecies of each individual plot.

*3.3. Distribution of Species*

No ubiquitous species, present in both the reserves, were found and no species widespread in Otonga was sampled (Supplementary Material Table S2). On the contrary, the species "n5" was found in all the plots in Otongachi. Regarding the forest types in Otonga (primary vs. secondary), 36 species were found in the primary forest only, 23 in the secondary forest only and 26 were shared between the two forest types (Figure 8).

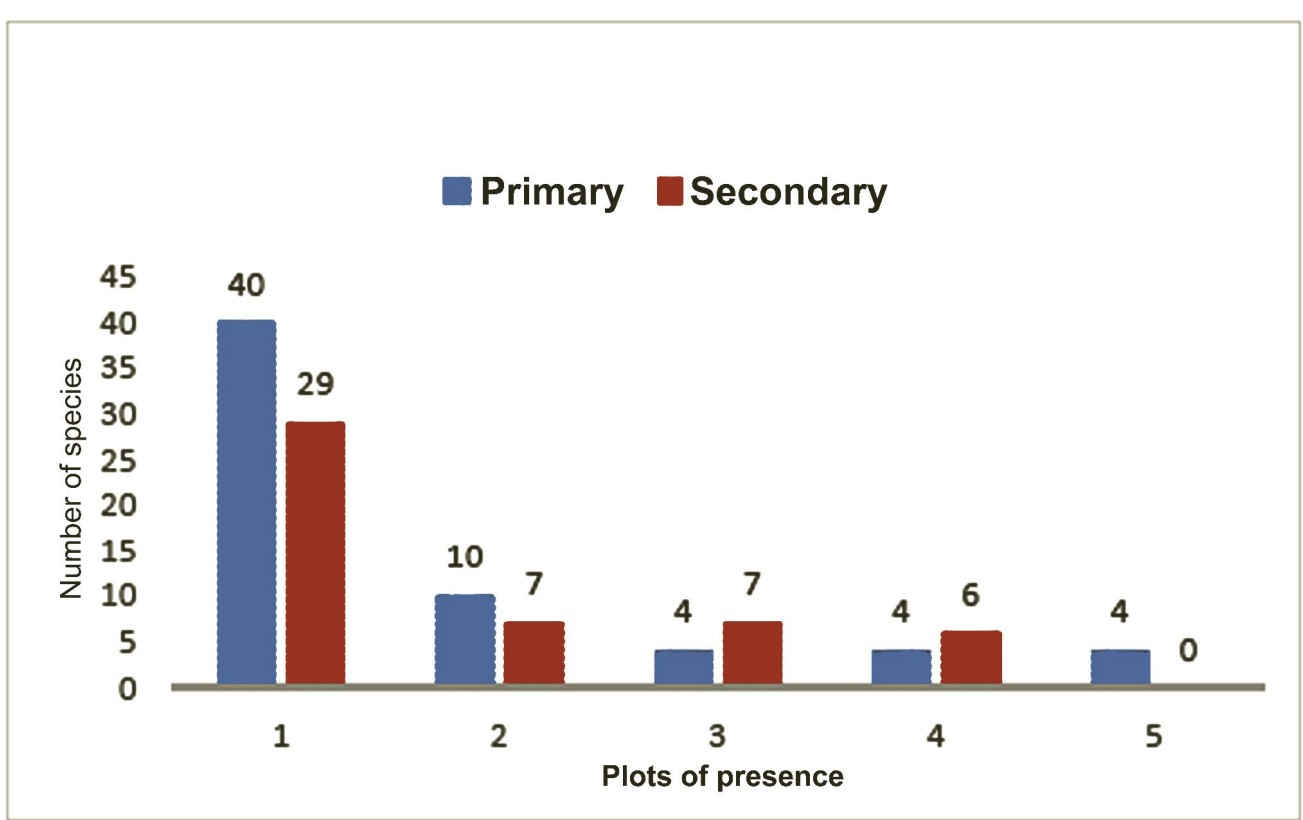

**Figure 8.** Abundance of species in relation to the number of plots where they have been collected. Otonga, primary and secondary forest.

This comparison was not possible in Otongachi, where there is only a secondary forest. The data gathered in both reserves reveal a large number of species for which only a few specimens were collected (rare_num). The number of singletons or doubletons is high: 50 singletons and 18 doubletons were found out of the 100 species collected in total (68% altogether). These data are different when referring to Otonga or Otongachi singularly: in Otongachi, 10 singletons and 2 doubletons out of 15 species were found (80% altogether), whereas in Otonga 40 singletons and 16 doubletons out of the total 85 species were sampled (66% altogether). In the primary forest, 22 singletons and 8 doubletons out of 36 species (83% altogether) were exclusive to this habitat. These species are distributed throughout the entire elevation range, with a slight increase towards the higher elevation; 17 of them were found in the two plots located above 2100 m; in the plot located at the highest elevation, "e280", 12 species in singletons or doubletons were found. Only 3 out of the 26 species that were present in both primary and secondary forest at Otonga were doubletons ("f1", "i4", "n2"). The 23 species found only in the secondary forest of Otonga were either singletons (18) or doubletons (5) (Figure 9).

The most abundant species found in the two reserves was "n5", which is ubiquitous in Otongachi and of which 87 specimens were found. There are no other taxa whose abundance or distribution is remarkable in this reserve: 11 out of the 15 species collected here were found only in one plot (Figure 10).

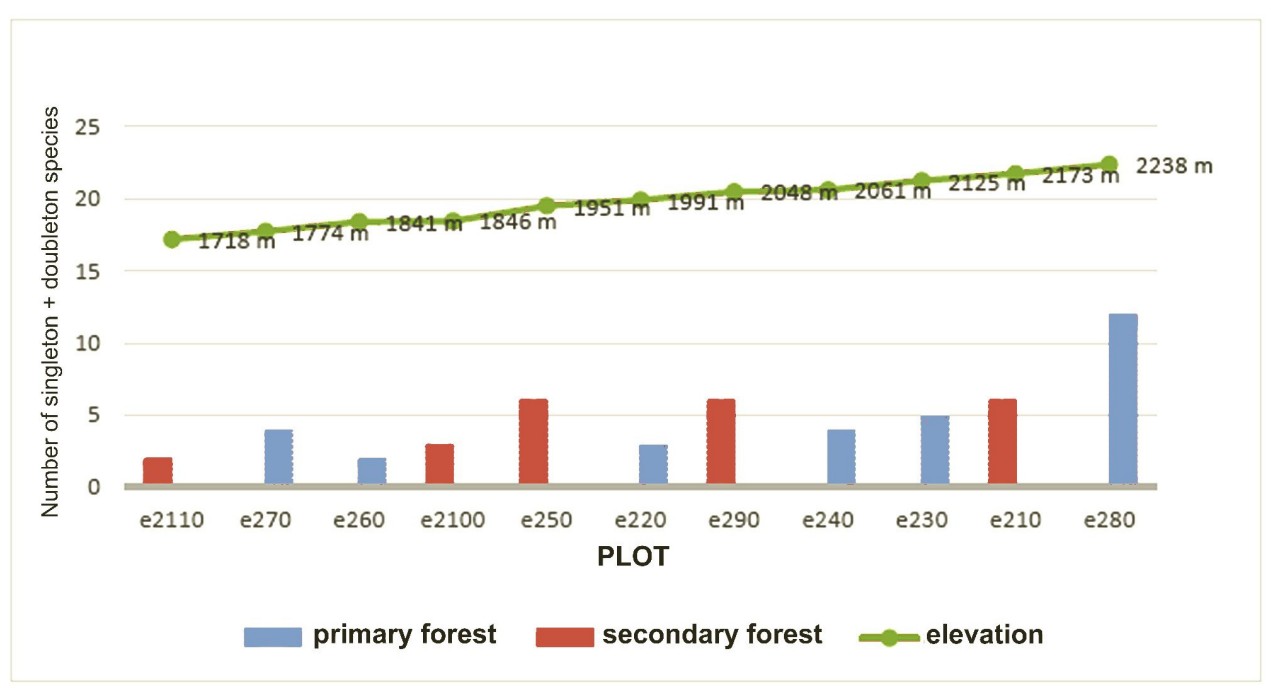

**Figure 9.** Abundance of singleton + doubleton species in Otonga primary and secondary forest.

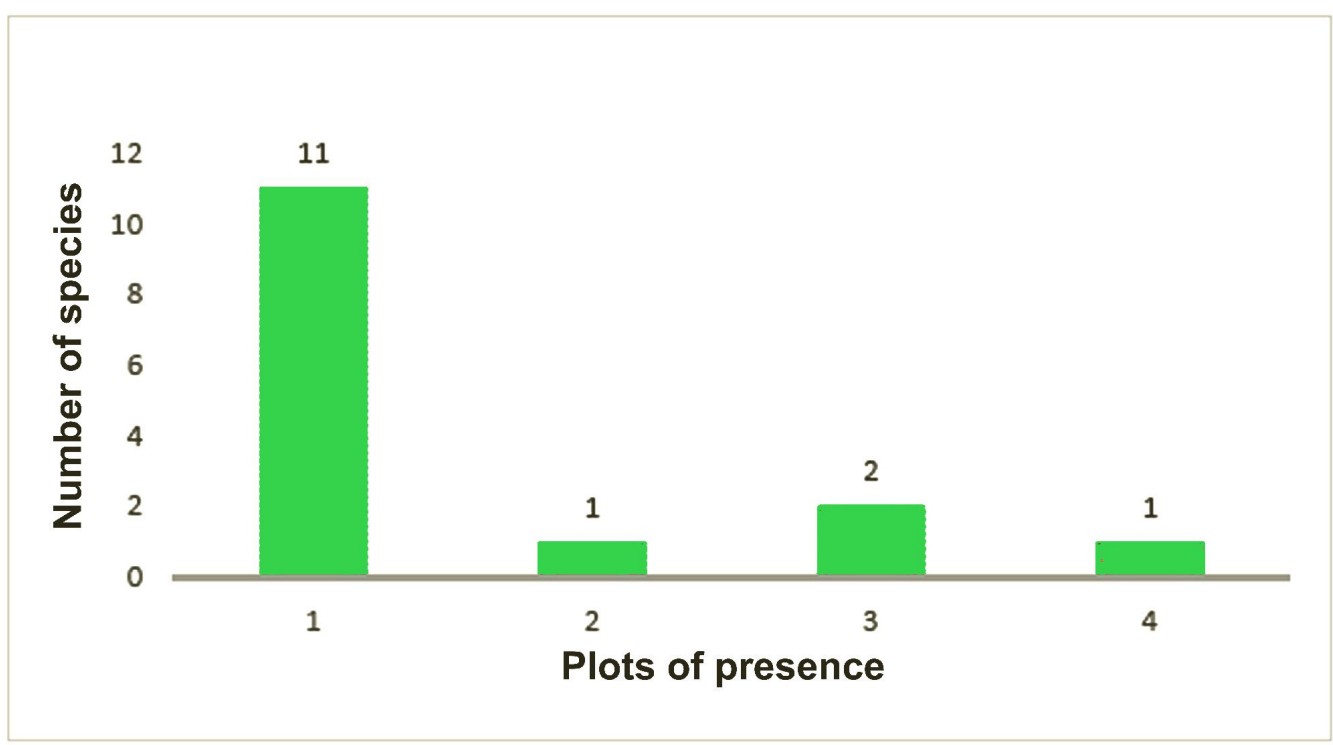

**Figure 10.** Abundance of species in relation to the number of plots where they have been collected. Otongachi, secondary forest.

Comparison of the abundance and distribution of the species collected in the different forest types was only possible in Otonga, given the absence of primary forest in Otongachi. There are a few species relatively rich in specimens in Otonga, but all of them have fewer specimens than "n5": the commonest one is "o5" (34 specimens), which is present in 9 out of the 11 plots (five in primary and four in secondary forest). The second most abundant species in Otonga is the related "o4", with 25 specimens from the same nine plots as "o5".

Other relatively abundant species in Otonga are "a2", with 27 specimens from seven plots, followed by "p2", with 26 specimens from eight plots. These two species are sympatric in the seven plots where "a2" is present. Another species relatively frequent in Otonga is "h5", found in 20 specimens from eight different plots. None of the 36 species found only in the primary forest was particularly abundant or widespread: "p1-1", "e6" and "i3" consisted of four specimens each, whereas in this forest type the most widespread species were present in no more than three plots each ("h3" and "i3"). The same trend was recorded for the 23 species exclusive to the secondary forest, all singletons or doubletons, and with only one species present in two plots ("k6").

*3.4. IndVal Species*

Through the application of the IndVal method, 14 species were significantly associated with a specific grouping of sites (Table 2).

**Table 2.** Species significantly associated with combinations of reserve/forest type.

| Species | Group | IndVal | *p*-Value |
|---|---|---|---|
| h3 | Otonga primary | 0.71 | 0.03 |
| i3 | Otonga primary | 0.71 | 0.03 |
| n5 | Otongachi secondary | 1.00 | <0.01 |
| d1 | Otonga secondary | 0.83 | <0.01 |
| m1 | Otonga secondary | 0.75 | 0.03 |
| o4 | Otonga | 0.91 | <0.01 |
| o5 | Otonga | 0.91 | <0.01 |
| a2 | Otonga | 0.85 | 0.01 |
| h5 | Otonga | 0.85 | <0.01 |
| h4 | Otonga | 0.80 | 0.02 |
| i2 | Otonga | 0.80 | 0.02 |
| k3 | Otonga | 0.80 | 0.02 |
| p2 | Otonga | 0.80 | 0.02 |
| k2 | Otonga | 0.74 | 0.03 |

Two species were exclusively associated with the Otonga primary forest, "h3" and "i3" (both with IndVal = 0.71, *p*-value = 0.03), while two were found to be associated with the Otonga secondary forest, "d1" (IndVal = 0.83, *p*-value = < 0.01) and "m1" (IndVal = 0.75, *p*-value = 0.03). One species, "n5" (IndVal = 1, *p*-value < 0.01), was associated with Otongachi. Eight IndVal species were associated with Otonga as a whole, some of them with high IndVal values, such as "o4" and the related "o5" (IndVal = 0.91, *p*-value < 0.01).

The mean IndVal for groups of species gave a significantly higher value for Otonga as the entire weevil community (IndVal = 0.66, *p*-value = 0.04), compared to the Otonga primary forest only (IndVal = 0.45, *p*-value = 0.02) and secondary forest only communities (IndVal = 0.49, *p*-value = 0.02) (Table 3).

**Table 3.** Weevil communities associated with combinations of reserve/forest type.

| Groups | Mean IndVal | *p*-Value |
|---|---|---|
| Otonga primary | 0.45 ± 0.02 | 0.02 |
| Otongachi secondary | 0.45 ± 0.05 | 0.02 |
| Otonga secondary | 0.49 ± 0.02 | 0.02 |
| Otonga (primary + secondary) | 0.66 ± 0.04 | 0.04 |

### 3.5. Community Composition

The sampling stations were represented in a scatter plot, in which the two axes of the graph represent the first two principal components extracted from the source data matrix (explained variability, PCoA1 = 12.34%, PCoA2 = 10.21%), where the distance between communities in the plots was calculated using the Jaccard index. Considering the abundance data, a clear distinction appeared in the scatter plot between Otonga and Otongachi in the distribution of samples along the first axis. Furthermore, the confidence interval (delimited in the graph by the dotted line) was larger for Otonga, revealing a greater variability in terms of community composition in Otonga than in Otongachi. Most of the plots in Otonga diverged markedly from the centroid, outside the 95% confidence interval, the most distant being "e260" and "e2100". The plots in Otongachi were much less divergent, with only a few plots outside the 95% confidence interval (Figure 11).

**Figure 11.** Scatter plot diagram of abundance data. The empty circles represent the Otonga sampling plots (right of diagram), the empty triangles the Otongachi sampling plots (left of diagram), the red dots represent the centroids. The axes PCoA1 and PCoA2 represent the first two principal components, while the dashed lines represent the 95% confidence interval.

A similar pattern was also evident in the scatter plot regarding presence/absence data: again, there was a clear distinction between Otonga and Otongachi in the distribution of the samples along the first axis (Figure 12).

The mean value of the distance of each plot from the centroid in the dispersion analysis was significant when considering abundance data (F-value = 12.16; *p*-value < 0.01). The mean value was 0.48 for Otonga and 0.33 for Otongachi (Figure 13).

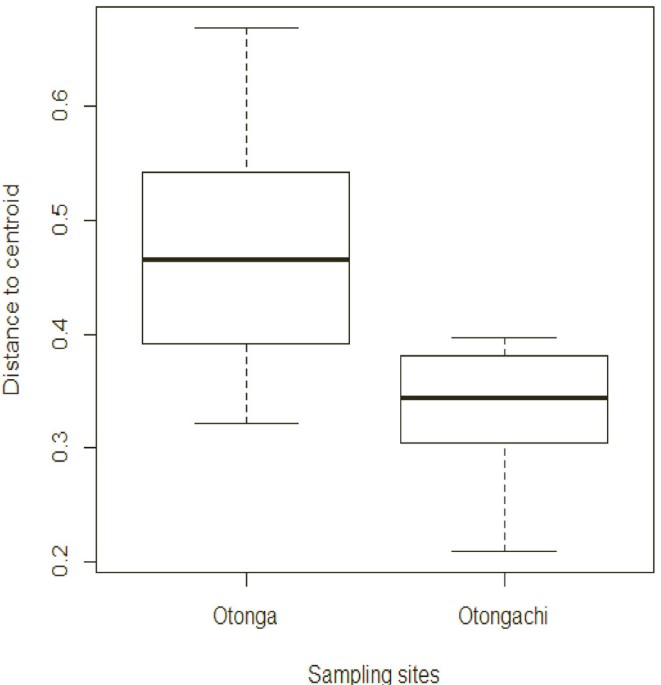

**Figure 12.** Scatter plot diagram of presence/absence data. The empty circles represent the Otonga sampling plots (right of diagram), the empty triangles the Otongachi sampling plots (left of diagram), the red dots represent the centroids. The axes PCoA1 and PCoA2 represent the first two principal components, while the dashed lines represent the 95% confidence interval.

**Figure 13.** Distribution of the abundance of species in the plots of Otonga or Otongachi with respect to the distance from the centroid identified by PCoA analysis.

The trend was similar for the presence/absence data, but in this case the differences in the dispersion around the centroid were not significant (F-value 1.18, *p*-value = 0.29).

The PERMANOVA analysis was carried out by first taking into account the abundance data of plots belonging to either the Otonga Reserve or the Otongachi Reserve. Significant differences were observed between the two reserves ($R^2 = 0.46$, *p*-value < 0.01). The PERMANOVA analysis of the presence/absence data produced similar results ($R^2 = 0.46$, *p*-value < 0.01). When this analysis was applied to plot groupings to test for differences between primary and secondary forest, no significant values were found for both abundance data (*p*-value = 0.19) and presence/absence data (*p*-value = 0.16).

The main factors influencing the composition of the weevil communities sampled in the various transects were determined by partitioning the variability in community composition according to the various categories of environmental variables measured. The percentage of variability explained by the model consisting of the variables considered in the presence/absence data is over 22%, a relatively high value, particularly in view of the inherent variability of animal communities. Most of the variability was determined by the area factor (Otonga or Otongachi), when considered as a function of the other variables. The second element in terms of explanatory importance was the elevation component (which has a value very close to that of the area factor), when considered as a function of the other variables; it occupies the first position when considered as a whole. The forest category (primary or secondary) loses almost all explanatory power when conditioned to the other variables (Table 4).

**Table 4.** Percentage of variability explained by the environmental variables.

| | | Weevil Presence/Absence | Weevil Abundance |
|---|---|---|---|
| Elevation | EXP | 23.81 | 26.6 |
| | PURE | 0.73 | 1.91 |
| Forest | EXP | 7.28 | 13.4 |
| | PURE | 0 | 1.05 |
| Area | EXP | 24.11 | 27.11 |
| | PURE | 0.46 | 1.85 |

The variability partition data for the previous categories were corrected and rescaled to assess their influence on the abundance data; no significant differences from the previously obtained ranking were observed. The other environmental variables identified by the habitat category (derived from rotten_log, moss_index, dryness_index and canopy) were not considered in this comparison because the variability explained by these was found to be low and not significant. A more detailed comparison between the habitat components given by the type of forest (primary or secondary), the category identified by the area and the elevation made it possible to identify the relative value of these individual components for the model consisting of the variables considered on the presence/absence data. All these components showed overlap. The area component explained almost as much variability as the elevation component, and both of these components are more explanatory than the forest component (Figure 14).

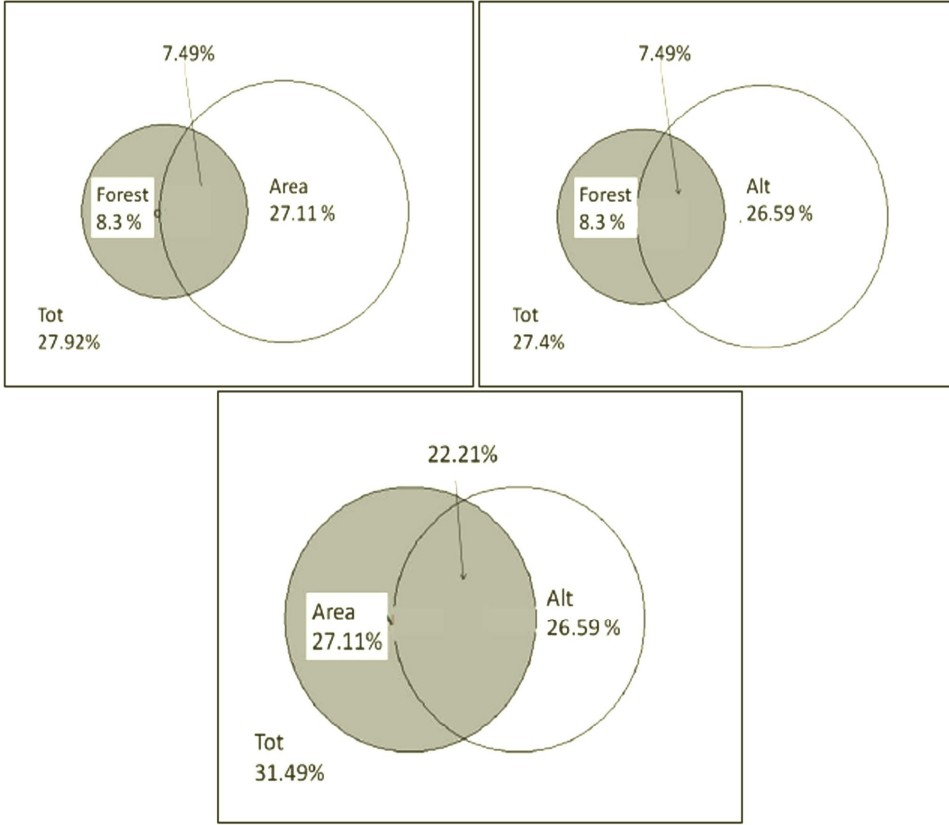

**Figure 14.** Percentage of variability explained by the pairs of components indicated, in order to identify the shared part of variability. Forest = primary or secondary forest, Alt = elevation levels, Area = Otonga or Otongachi. Model built on presence/absence data.

A very similar result was obtained by comparing the relative values of the same components for the model built on the abundance data.

## 4. Discussion

This is the first study that took into the litter weevil fauna of tropical forests in South America, comparing different types of forests (specifically, tropical montane cloud forest and foothill evergreen forest), primary and secondary forests and different elevation levels. The richness and density of weevils found in the Otonga (TMCF) and Otongachi (FEF) reserves during the study, 510 specimens belonging to 100 morphospecies, included in roughly 24 morphogenera, were notable, particularly considering the limited number of collecting sites and the short time during which the study was carried out. Possible variations in the composition of the communities at different times of year or during the relatively drier season could not be taken into account. However, even though we could not carry out a larger number of samplings, we are confident that the results of our research offer previously unavailable information about diversity of litter weevils of the Ecuadorian TMCFs and provide evidence of their role as bioindicators usable to characterise the biocoenoses of these tropical forests, helping to assess their environmental quality. According to Basset et al. [45], in fact, "to determine the species diversity of a tropical rainforest, the total area sampled need not be overly large—provided that the sampling design adequately covers both microhabitats and plant species".

The mean density of weevils was about five specimens per 3 L per each one-square-metre sampling point, which converts to about 50,000 per ha. Considering that the total surface of Otonga and Otongachi is about 1650 ha, this means approximately 80 million litter weevils in the two reserves, but only in 3 L of soil per square metre. In the plots that were sampled very often, the quantity of litter far exceeded 3 L per square metre,

therefore it can be very roughly, but reasonably, estimated that Otonga and Otongachi contain hundreds of millions of specimens, an amazingly rich fauna. A rough count of the numerosity of the litter weevils was also attempted in a Mexican forest [8], where 170,000 individuals per hectare were estimated. In this study, however, only 57 species, assigned to 10 genera, from over 2000 specimens sampled were identified (including a few species that appeared not to be true leaf litter obligate inhabitants, since they were winged). So, the Ecuadorian forests investigated host a much richer diversity, even though apparently the density of specimens is lower. As for the Mexican habitat, in Ecuador the vast majority of species were also undescribed.

The plots were selected so that a comprehensive example of the various coenoses could be investigated, with particular regard to the different elevation levels and forest types but, quite obviously, our sampling plots were undersampled, and only in a few cases did the collected species approach the expected richness. The communities in these ecosystems are apparently composed of a large number of species, the majority present in a small number of specimens, as suggested by the high percentage of singletons and doubletons, and rare species in general. A percentage of infrequent species, exceeding the expected, is a common finding when studying the arthropod fauna of tropical forests, not only for weevils, but for other arthropods associated with litter as well [45–48]. A more exhaustive sampling could not be applied to our study, mainly for practical reasons (research time and sorting equipment were limited). In any case, thanks to the relevant species richness of our samplings and its good correlation with the estimated one, it was possible to characterise the communities present in the two reserves, revealing the differences between them and assessing their association with the observed environmental variables. Otonga and Otongachi are colonised by distinct weevil communities, and thus can be considered as different coenoses. Otongachi differs from Otonga in the lower elevation (800–1000 m vs. 1700–2300 m), the type of forest, FEF instead of TMCF and EMF, the absence of primary habitats, the lower thickness of the litter and a much heavier anthropic impact.

Otonga hosts 85% of the species found in our study. A high fidelity to a single plot was detected, with a low number of species present in more plots. Almost no data on biotic and abiotic factors are available for each of the elevation levels, but it is possible that the distribution of some parameters (mean annual rainfall, cloud coverage, air and soil humidity, mean annual temperature, tree species, understory vegetation and more) varies to some extent in the different elevation levels. Elevation was found to be one of the most significant variables in determining the faunistic composition of the litter weevils. In Otonga, the species richness was found to increase from the lower to the higher elevation levels, with the highest number of species found in the plot at the top (32 species, plot "e280"), an area of primary forest with large trees and thick litter, with intermediate soil wetness, features that may have favoured the considerable number of weevils present. In this plot, the sampled species closely approached the number of estimated species. An increment of species towards higher elevations, within the range of presence of the TMCFs, was also observed in studies on different groups of arthropods [21]. Even though only morphospecies were used, these were assigned to provisional genera. It appears that, in some cases, closely related vicariant species were present at different elevation levels. Among the Molytinae: Anchonini, the two morphological, and probably systematical, vicariant taxa "n3" and "n4" were present, respectively, in the level ranging from 1700 to 1800 m and in the level ranging from 1800 to 1900 m, without any overlap. Additionally, for the two morphologically vicariant taxa "m7_1" and "m7_2", there is a separation according to the elevation. In this case, "m7_1" was found from 1700 to 1900 m, whereas "m7_2" was present in the 2000 to 2100 m level (Figure 15).

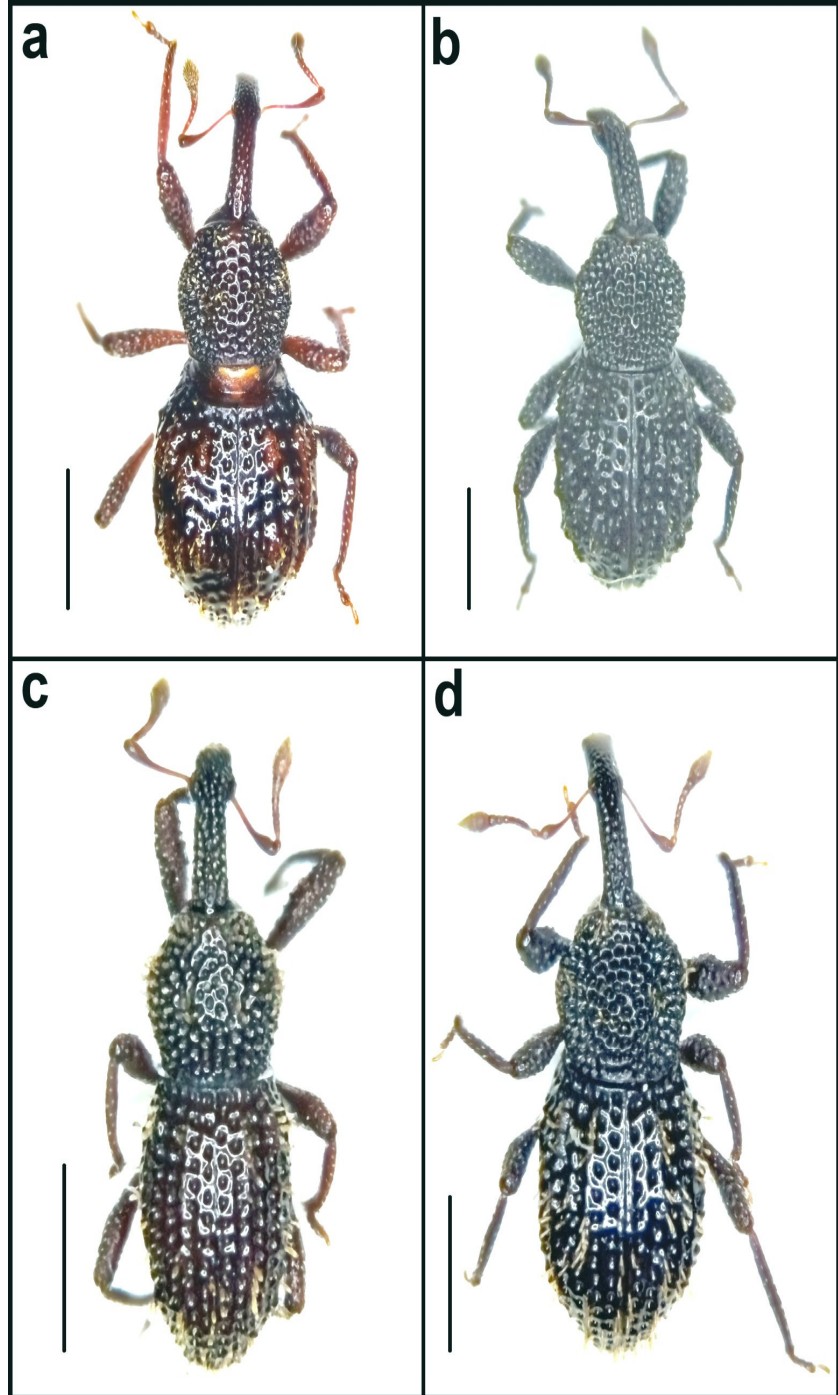

**Figure 15.** Weevils from Otonga. Species "n3" (**a**); species "n4" (**b**); species "m7_1" (**c**); species "m7_2" (**d**). Bar: 1 mm.

These observations, if confirmed after a taxonomic study, could indicate that these species underwent allopatric speciation at different elevations. This would confirm observations that were made for other weevil species from tropical forests, such as the extremely speciose genus *Trigonopterus* Fauvel, 1862, native to the tropical forests of South-East Asia, recently investigated [49–52]. This finding is also in agreement with the high β-biodiversity values demonstrated by other studies conducted in the TMCFs [53] and with biogeographic models, according to which these forests are characterised by a high rate of allopatric speciation [8]. An allopatric speciation that occurred in spatially or altitudinally close areas indicates that these are very highly specialised elements, strictly stenoecious, that seem

to have adapted to a very narrow range of environmental variation and have a very low capability of dispersion or adaptation to modifications of their habitat. Their presence implies that the habitat did not suffer from extremely severe anthropic impacts, which these taxa could not tolerate. There is another observation suggesting that the TMCF of Otonga, probably thanks to the efforts of the Otonga Foundation, remained well preserved in the areas where the primary forest was impacted to some extent and is now replaced by secondary forest. We evidenced only limited differences between the primary and secondary forests in the composition of the weevil communities. The overall numbers of species and specimens were quite similar (mean number of species per plot 18.0 ± 3.7 with 36.5 ± 10.0 specimens in the primary and 17.4 ± 7.5 species with 34.8 ± 7.5 specimens in the secondary forest), and only four, out of 85 species found in Otonga, were statistically associated with Otonga primary ("h3" and "i3") or secondary forest ("d1" and "m1") (Figure 16).

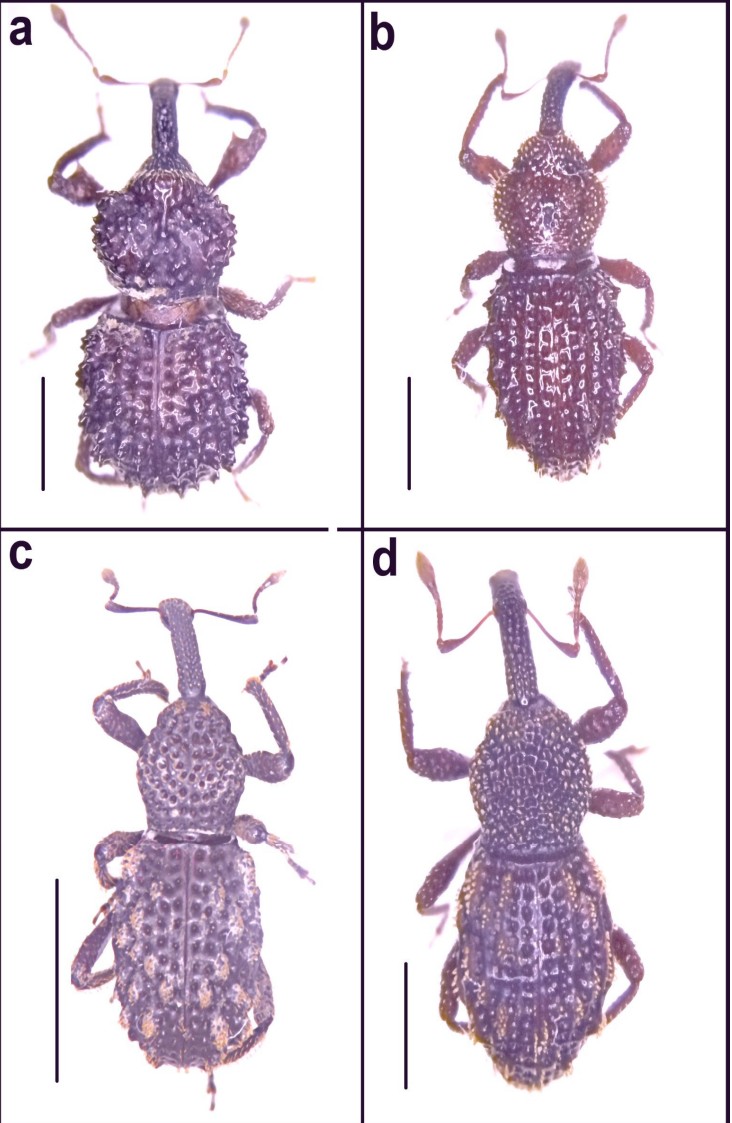

**Figure 16.** Weevils from Otonga. Species IndVal of the primary forest. Species "h3" (**a**); species "i3" (**b**). Species IndVal of the secondary forest. Species "d1" (**c**); species "m1" (**d**). Bar: 1 mm (**a,b,d**); 5 mm (**c**).

The species richness was similar between the two types of forest, and only the singletons and doubletons were more frequent in the primary patches. The blind species of

morphogenus "p" give some more precise hints in the evaluation of the Otonga forests. These weevils are closely associated with the deeper part of the litter, require very specific and stable environmental conditions and have low mobility. As expected, they were mainly sampled in primary forests, but one species, namely, "p2", was also present in two plots in secondary forests ("e210" and "e250"). At least in these plots, the habitat must have suffered a limited impact when the primary forest was replaced by a secondary forest, indicating that this succession occurred gradually, and that the primary forest was not completely removed and replaced by a uniform grassland, but at least some patches with forest remained untouched. A colonisation of the secondary forest that occurred after the time of its planting (less than 30 years ago) is in fact extremely unlikely. Quite interestingly, in one of these two plots of secondary forest ("e250"), 23 species were found, making it the second richest sampling site of Otonga. It is not so unusual that in secondary forests a high diversity can be detected. The litter fauna in the primary, undisturbed patches of TMCF appears to be well structured, quite stable for a long time and mostly composed of numerous strongly specialised species, most of which are present in a low number of specimens. A large part of this primary fauna could survive in plots of secondary forest that did not suffer from a particularly intense level of anthropic impact, but these communities may also include some species more tolerant and resilient to human disturbance and with a higher dispersion capability, which have been able to colonise the patches from more deteriorated habitats. The presence of abundant soil arthropods in secondary, or restored, forests was also detected in other cases [54,55], but in these studies, even though the soil arthropods were abundant, quite often the species composition was quite different from that of the primary forests, and in general it was dependent on the distance from the patches of untouched vegetation [54]. Moreover, these studies did not specifically evaluate the litter weevils, which, including many specialised taxa, offer a more accurate response.

Despite the fact that no quantitative data on litter thickness were collected during the current study, random observations of the depth of litter present in the vicinity of the plots showed that the thickness was consistently greater in Otonga than in Otongachi. This is likely to be one of the most important factors responsible for the much more abundant weevil fauna present in Otonga, even though more habitat constraints may have contributed to determine the different aspects of the communities. When forest litter is well structured and quite deep, its conditions are more stable, an essential requirement for most species. The low number of species in the plot "e260", compared with the very speciose nearby plots "e270" (primary forest) and "e250" (secondary forest), appears to be determined by the unsuitable conditions of the litter, whose structure is deficient due to its low thickness and the quite superficial rock layer, and is therefore subject to unstable conditions; food availability in shallow litter may be a limiting factor and water content can rapidly vary from dryer to wetter conditions. Intermediate humidity conditions were found in the plots with the highest number of species, and these conditions are granted by a thicker layer of litter, that acts as a buffer to stabilise humidity, and contains a much higher availability of decaying material. The presence of larger dead branches and logs did not significantly determine a variation in the species richness; most species, at the larval stage, are xylosaprophagous (such as the Molytinae: Anchonini)—but Raymondionyminae feed on roots. However, the adults can move around, feeding on smaller branches and small fragments of decaying organic matter at some distance from the logs.

As said, no species were present in both Otonga and Otongachi forests, despite the limited distance between the areas. This was not completely unexpected, since it is known that the spatial and ecological variation of the soil arthropods is particularly consistent, not only in the tropics but also in other regions [45,56]; moreover, the habitat in Otongachi has been severely impacted by anthropic disturbance, much more than in the secondary forests of Otonga. Comparisons between the two reserves give further details for evaluating the differences in the communities between well-preserved and severely disturbed secondary habitats. In particular, the presence of a dominant species in Otongachi, "n5", widespread in all the plots and accounting for 71% of the specimens collected and the other taxa being

found only in singletons or doubletons are remarkable. The presence of one, or at most very few, dominant species seems to be a typical feature of anthropogenically influenced and/or unsuitable ecosystems ([57], and references therein).

The most sensitive and highly adapted species are not able to re-colonise recovered habitats after disturbance, so the ideal management would be to leave these habitats untouched, but anyway constantly checked to guarantee that the ecological conditions do not vary. However, the few differences that were detected between the Otonga primary and secondary TMCFs indicate that a limited disturbance can be tolerated, if proper management is implemented and special attention is paid to keeping the litter layer as unaltered as possible. Even if not including a tropical forest, a survey on litter weevils in Mexican oak forests with different disturbance regimes showed a correlation with the disturbance level regarding species richness and abundance, and concluded that, similarly to what we have observed in Otonga, " . . . leaf litter weevil communities in oak forests of central Mexico are similar in taxonomic composition and richness to cloud forests from southern Mexico and that even small, moderately disturbed fragments may be sufficient to maintain their populations" [58]. The poor and highly altered communities found in Otongachi show that severe anthropic impact can render the forest unsuitable for the most specialised taxa. Only one tolerant species found proper conditions in this area, and the others, probably more specifically adapted to this low-elevation forest, are extremely rare and scattered, and some may have disappeared. In order to achieve a more complete understanding of the causes that determined the imbalance of the community of Otongachi, more research is needed in areas intermediate between Otonga and Otongachi, and in plots characterised by apparent good ecological conditions.

Litter weevils are demonstrated to be extremely useful bioindicators, and, in general, the study of these specialised litter arthropods is confirmed to be an indispensable tool in bringing detailed information on the biocoenoses in natural and altered tropical forests and for evaluating the results of forest management, not differently from what can be achieved from other taxa, such as ants [59]. Information that can be recovered from their analysis can be used to refine data obtained from the entire weevil fauna, whose $\beta$-diversity can be quite high even in "artificial" environments, such as oil palm plantations [60]. In these cases, a comparatively rich diversity is determined by spatial turnover of many non-specialised elements. Knowledge of the range of the more specialised litter weevils of the montane tropical forest is almost absent, but a few revisory studies on single genera [49–52,61] suggest that in many instances most taxa have a very narrow range, even limited to a single location. In the case of Otonga, we cannot exclude that some of the species, i.e., those that were found only at a single elevation level and are replaced by vicariant species at different elevation levels, are exclusive to this region and perhaps even limited to a very narrow area. These species are therefore intrinsically at risk, either as a consequence of anthropic impact or possibly also in the case that climate change modifies some characteristics of the litter, with particular regard to the humidity balance.

A critical use of morphospecies can overcome the taxonomic impediment so frequently encountered when studying tropical fauna. However, it cannot be forgotten that only with complete knowledge of the taxa and recognition of their phylogenetic relationships can a more precise evaluation of the habitats investigated be obtained, and the interrelations between them be disclosed. Moreover, cataloguing the diversity is essential to increase our knowledge of these habitats and assess conservation needs, particularly in view of the environmental changes that are rapidly occurring.

The data on the litter weevil coenoses of Otonga and Otongachi can have a broader application to other sites of TMCF, and tropical montane forests in general. In conservation strategies, quite often large vertebrates are well-known symbols and are used as indicators of habitat quality; they are also suitable to spread information outside the scientific community about the necessity of providing good protection to what remains of natural habitats in the tropics. Most vertebrates, however, can move and look for proper conditions, within the range of each species. These highly specialised arthropods are incapable of any

significant movement, and the destruction of their habitats, or a strong impact on them, will inevitably result in their extinction. This is probably what has occurred in Otongachi, whereas full diversity seems to remain in Otonga and must be accurately conserved.

**Supplementary Materials:** The following supporting information can be downloaded at: https://www.mdpi.com/article/10.3390/d14100871/s1, Table S1: List of the sampling stations. Table S2: List of the morphospecies with distribution in the plots.

**Author Contributions:** Conceptualisation, O.M., C.C. and M.M.; methodology, C.C., C.B. and M.M.; field research, O.M.; morphospecies identification: M.M.; data analysis, C.C.; writing—original draft preparation, O.M.; writing—review and editing, M.M., C.C. and C.B. (current knowledge on the litter weevils of the Otonga region). All authors have read and agreed to the published version of the manuscript.

**Funding:** This research received no external funding.

**Acknowledgments:** We would like to thank Giovanni Onore, President of the Fundación Otonga and founding member of WBA Onlus, for his kind help in Ecuador, for his hospitality in Quito, in the Otongachi Research Centre and in the Otonga forest, for his support during the organisation and carrying out of the present research, for facilitation with the QCAZ Zoological Museum of the Catholic University of Quito and for his unceasing work to promote scientific knowledge and to protect the biodiversity of Ecuador. Kind help came from Cesar Tapia and his sons, Elicio and Italo, members of the Fundación Otonga, who gave thoughtful support during field work in the Otonga and Otongachi Reserves, and Vilman Tapia, who, with her fine tailoring, built the Winkler sorters used in this research. In WBA Onlus we found great sensibility and support in the form of the Gianfranco Caoduro, who encouraged and helped us during the organisation of this research.

**Conflicts of Interest:** The authors declare no conflict of interest.

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
