# Peer review of "Diversity and Spatial Distribution of Leaf Litter Curculionidae (Coleoptera: Curculionoidea) in Two Ecuadorian Tropical Forests"

_diversity, doi:10.3390/d14100871_

Round 1

Reviewer 1 Report

The manuscript entitled Diversity and spatial distribution of leaf litter Curculionidae (Coleoptera: Curculionoidea) in two Ecuadorian tropical forests (Ecuador) is well-written and explores the diversity of weevils in two different forests. The authors studied species abundance and composition at different elevation gradients and forest composition. This manuscript could be accepted for publication after a minor review.

Below are some minor comments:

Lines 64-77 – Suggest moving this paragraph to the end of the introduction.

Lines 90-91 – Make sure the format style is correct. Also, the following two paragraphs could be combined.

Figure- Legend (Leyenda) and its content have not been translated into English.

Table 5 – In the AREA row- Was it supposed to be Exp and Pure?

Author Response

Thank you very much for your positive comments!

We have corrected the text according to your suggestions. Lines 90-91 were indeed not precise, the first line is the title of a subchapter.

Massimo Meregalli

Reviewer 2 Report

The research addressed the main question about"Is weevils within the litter community of tropical forests a possible indicator taxon of the biodiversity?". It’s pretty interesting and original for the large number of data collected in relation to the very rich fauna of Curculionidae in the soil of Reserves. It establishes the number of weevils specimens in three liters of soil and per hectare by calculating the species already known and even those yet to be described. The paper is well written and easy to read but dedicated to specialists in the subject with adequate statistical preparation. Also, The conclusions are consistent with the evidence presented, it is significant that about 80 million weevils have been calculated in the two reserves of Otonga and Otongachi. It gives a good idea of the impact of this taxonomic group on the environment for the maintenance of biodiversity in the rainforest of Ecuador.

There were only minor typos. For the rest, the manuscript is very well structured and must be considered a milestone in the study of the arthropods of the equatorial forest litter. With my compliments

PD

Author Response

Thank you SO MUCH!! for your comments! It's not usual to have a reviewer saying that the paper is a milestone, and we all were extremely pleased to read that our efforts have been appreciated and the paper is considered useful for the research on tropical forest Arthropods. We have checked the text and corrected the typos.

Massimo Meregalli